

**Persistent primary organic tar particles during the regional wintertime hazes in**
**North China: insights into their aging and optical changes**
Lei Liu[1], Jian Zhang[1], Yinxiao Zhang[1], Yuanyuan Wang[1], Liang Xu[1], Qi Yuan[1], Dantong Liu[1], Yele
Sun[2], Pingqing Fu[3], Zongbo Shi[4], and Weijun Li[1,*]
[1]Department of Atmospheric Sciences, School of Earth Sciences, Zhejiang University, Hangzhou
310027, China
[2]State Key Laboratory of Atmospheric Boundary Layer Physics and Atmospheric Chemistry, Institute
of Atmospheric Physics, Chinese Academy of Sciences, Beijing 100029, China
[3]Institute of Surface-Earth System Science, Tianjin University, Tianjin 300072, China
[4]School of Geography, Earth and Environmental Sciences, University of Birmingham, Birmingham
B15 2TT, UK
*Correspondence to*: Weijun Li (liweijun@zju.edu.cn)





**Abstract**
Primary organic aerosol (POA) is a major component of $PM_{2.5}$ in the winter polluted air in the
North China Plain (NCP), but our understanding on the atmospheric aging process of POA particles
and the resulting influences on their optical properties is limited. As part of the Atmospheric Pollution
and Human Health in a Chinese Megacity (APHH-Beijing) programme, we collected airborne
particles at an urban site (Beijing) and an upwind rural site (Gucheng, Hebei province) in the NCP
during 13–27 November 2016 for microscopic analyses. We confirmed that a distinct group of
spherical or irregular POA particles with high viscosity, defined as primary organic tar (POT) particles,
was emitted from the domestic coal and biomass burning at the rural site and was further transported
to the urban site during the regional wintertime hazes. During the heavily polluted period ($PM_{2.5}$ >
200 μg $m^{-3}$), more than 60% of the POT particles were thickly coated with secondary inorganic
aerosols (named as core–shell POT-SIA particle) through the aging process, suggesting that POT
particles can provide surfaces for the heterogeneous reactions of $SO_2$ and $NO_x$. As a result, their
average particle-to-core ratios at the rural and urban sites in the heavily polluted period increased to
1.60 and 1.67, respectively. Interestingly, we found that the aging process did not change the
morphology and sizes of the POT cores, indicating that POT particles are quite inert in the atmosphere
and can be transported long distances. We using the Mie theory estimated that the light absorption of
individual POT particles was enhanced by ~1.39 times in the heavily polluted period at the rural and
urban sites due to the lensing effect of secondary inorganic coatings. We highlight that the lensing
effect on POT particles should be considered in radiative forcing models and the governments should
continue to promote clean energy in rural areas to effectively reduce primary emissions.



## 1 Introduction

Atmospheric aerosol particles can affect the regional and global energy budget by scattering and absorbing solar radiation, modify the microphysical properties of clouds by acting as cloud condensation nuclei (CCN) and exert adverse effects on human health such as respiratory and cardiovascular diseases (IPCC, 2013; West et al., 2016). With the rapid industrialization and urbanization in past decades, severe air pollution characterized by high concentrations of fine particulate matter ($PM_{2.5}$) frequently occurs in China, especially the regional hazes in the North China Plain (NCP), which has received wide concerns from the public, governments, and scientists (Sun et al., 2016). Many previous studies have well documented that the synergetic effects from extensive emissions of primary particles and gaseous precursors, efficient secondary aerosol formation, regional transport, and unfavorable meteorological conditions are the main factors contributing to the haze formation in the NCP (Chang et al., 2018; Liu et al., 2016; Zhong et al., 2019). In particular, the long-term measurements have confirmed that the wintertime haze episodes in Beijing are commonly initiated by regional transport of air pollutants from the south parts of NCP (e.g., Hebei and Henan provinces) under weak southerly winds and then evolved through the massive secondary aerosol formation via heterogeneous reactions (Ma et al., 2017; Sun et al., 2014; Zheng et al., 2015).

During the regional transport and evolution of haze episodes, complex physical and chemical processes in the atmosphere, such as condensation, coagulation, and heterogeneous reactions, could largely alter the morphology, composition, size, and mixing state of individual particles, which is also known as "particle aging" (Li et al., 2016a). Particle aging could further influence the optical property, health effects, hygroscopicity, and CCN activity of aerosol particles, although different types of particles might have different impacts (Fan et al., 2020; Li et al., 2016b; Riemer et al., 2019). Up to now, most of the studies conducted in the NCP mainly have applied various bulk online and offline aerosol analytical techniques (e.g., online aerosol mass spectrometer (AMS) and offline ion chromatographer (IC)) to explore mass concentrations, possible sources, and formation mechanisms



of different aerosol components, such as sulfate, nitrate, and organic aerosols (Chen et al., 2020;
Cheng et al., 2016; J. Li et al., 2020; Sun et al., 2016; Wang et al., 2020). However, the knowledge
on the aging process of aerosol particles remains limited. Therefore, it is of great importance to
document the aging processes of different particles, which can reveal the particle transformation in
the atmosphere and better assess the climatic effects of aerosols (Du et al., 2019; Li et al., 2016a).

Field observations have shown that carbonaceous aerosols, including organic aerosol (OA) and

black carbon (BC), are the dominant component of $PM_{2.5}$ during the heating season in the NCP, which
accounts for more than 50% of the total $PM_{2.5}$ mass concentration (Liu et al., 2020; P. Liu et al., 2017;
Zhang et al., 2020). Source apportionment results reveal that residential coal and biomass burning in
rural areas are the major contributors to the carbonaceous aerosols during the wintertime haze in the
NCP (Li et al., 2017). BC is the major light-absorbing aerosol in the atmosphere, which can strongly
absorb solar radiation and thus affect the regional and global climate (Bond et al., 2013; D. Liu et al.,
2017; Wang et al., 2014). In recent years, a bunch of studies have well documented the aging process
of BC particles and revealed that the secondary inorganic and organic aerosol coatings (e.g., sulfate
and organics) can significantly enhance the light absorption capacity of the internally mixed BC
particles via the "lensing effect" (Chakrabarty and Heinson, 2018; Y. Wang et al., 2017). Recently,
light-absorbing organic aerosols, also known as brown carbon (BrC), has been reported to be
ubiquitous in the atmosphere in the NCP (Wang et al., 2018; Xie et al., 2019). Many studies have
demonstrated that primary OA (POA) emitted from the residential coal and biomass burning is the
major source of BrC and the chemical composition and optical properties of BrC in freshly emitted
POA, as well as the BrC in the ambient atmosphere, were analyzed in detail using bulk techniques
such as mass spectrometry and UV–visible spectrophotometry (M. Li et al., 2019; X. Li et al., 2020;
Song et al., 2018; Sun et al., 2017; Yan et al., 2017). However, only a few studies characterized the
microscopic information such as the morphology and mixing state of freshly emitted POA particles
by transmission electron microscopy (TEM) (L. Liu et al., 2017; Zhang et al., 2018). The abundance





and the aging process of POA particles in the atmosphere and the resulting influences on their optical
properties remain unknown in the NCP.

This study, as part of the Atmospheric Pollution and Human Health in a Chinese Megacity

(APHH-Beijing) programme (Shi et al., 2019), aims to explore the atmospheric aging process of POA
particles emitted from the residential coal and biomass combustion following the regional transport
and evolution of haze episodes. Individual particle samples were collected in urban Beijing and the
surrounding rural regions during the winter campaign and then were analyzed by microscopic
methods to obtain the morphology, composition, size, and mixing state of different individual particle
types. Besides, the bulk analyses of aerosol chemical components were also conducted to help
understand the evolution of haze episodes. We found that a distinct group of POA particles emitted
from the domestic coal and biomass burning were abundant in winter in the NCP. For the first time,
we characterized the aging process of such POA particles based on microscopic analyses and Mie
theory was used to further explore the resulting influences on their optical properties.

**2 Experimental methods**

**2.1 Sampling sites and sample collections**

Field observations were carried out simultaneously at the Beijing (BJ) urban site (39°58'27" N,

116°22'16" E) and Gucheng (GC) rural site (39°08'58" N, 115°44'00" E) during 13–27 November
2016. The locations of two sampling sites in the NCP are displayed in Fig. 1a. The BJ urban site,
located on the rooftop of a two-story building (8 m above ground level (a.g.l.)) in the Tower Division
of the Institute of Atmospheric Physics, Chinese Academy of Sciences, is between the north 3rd and
4th ring road and surrounded by commercial area and residential apartments (Fig. 1b). The GC rural
site, located on the rooftop of a three-story building (12 m a.g.l.) at the Gucheng Integrated
Ecological–Meteorological Observation and Experimental Station of the Chinese Academy of
Meteorological Sciences in Dingxing county, Hebei province, is 120 km to the southwest of the BJ
urban site and surrounded by many villages and farmlands (Fig. 1c). The detailed information about



the two sampling sites can be found in the introduction paper of the APHH-Beijing programme (Shi
et al., 2019). The 24-h backward trajectories of air masses ending at the height of 100 m (a.g.l.) over
the Beijing site (Fig. 1) were calculated using the NOAA Air Resources Laboratory's HYbrid Single-
Particle Lagrangian Integrated Trajectory (HYSPLIT) model (Stein et al., 2016).

At the BJ urban site, the species in non-refractory submicron aerosols (NR-PM$_1$) including

organic matter (OM), $SO_4^{2-}$, $NO_3^-$, $NH_4^+$, and $Cl^-$ were measured by a high-resolution aerosol mass
spectrometer (HR-AMS, Aerodyne Inc., USA). At the GC rural site, PM$_{2.5}$ samples were separately
collected during the daytime (8:30 am to 8:00 pm) and nighttime (8:30 pm to 8:00 am the next day)
onto 90 mm-diameter quartz fiber filters (Pallflex 7204, Pall Corporation, USA) using a medium-
volume sampler (TH-150A, Wuhan Tianhong Instruments Co., Ltd., China) at a flow rate of 100 L
min$^{-1}$ for 11.5 h. Field blank samples were collected for approximately 15 min without starting the
sampler. The filters were prebaked at 450°C for 6 h before sampling to remove any possible
contaminants. All the collected samples were sealed individually in aluminum foil bags and stored in
a refrigerator at −20 °C for further analyses.

Individual particle samples were collected onto copper (Cu) TEM grids coated by formvar and

carbon films (carbon type-B, 300 mesh, Beijing XXBR Technology Co., Ltd., China) at the GC rural
and BJ urban sites using an individual particle sampler (DKL-2, Qingdao Genstar Electronic
Technology Co., Ltd., China) at a flow rate of 1 L min$^{-1}$. The DKL-2 sampler consists of a single-
stage impactor with a jet nozzle of 0.5 mm in diameter. Sampling duration ranged from 8 s to 3 min
depending on the pollution levels to avoid overlap of particles on the TEM grids. Individual particle
samples were placed in a clean and airtight container with controlled temperature ($T$, 25±1°C) and
relative humidity (RH, 20±3%) for further analyses. The detailed information about the individual
particle samples collected at the two sampling sites is listed in Table S1.

Meteorological parameters including $T$, pressure ($P$), RH, wind speed (WS), and wind direction

(WD) were recorded every 5 min at two sampling sites using a pocket weather station (Kestrel 5500,



Nielsen-Kellermann Inc., USA). The hourly concentrations of PM$_{2.5}$ and gaseous pollutants (i.e., SO$_2$,
NO$_2$, CO, and O$_3$) during the sampling period at two monitoring stations (i.e., Dingxing government
station: 39°15'42" N, 115°48'06" E; Beijing Olympic center station: 40°00'11" N, 116°24'25" E) close
to GC rural and BJ urban sites were downloaded from the website of air quality online monitoring
and analysis platform (https://www.aqistudy.cn/). All the data in this study are presented at the Beijing
local time (UTC+8).
**2.2 PM$_{2.5}$ chemical analysis**

PM$_{2.5}$ samples collected at the GC rural site were analyzed to obtain their water-soluble inorganic

ions (WSII), organic carbon (OC), and elemental carbon (EC). For the analysis of WSII, two punches
with a diameter of 16 mm from each PM$_{2.5}$ sample were put into a vial, followed by adding 20 mL
deionized water (18.2 MΩ). Then these vials were placed in an ultrasonic water bath for 30 min to
extract WSII. The solutions were further filtered using PTFE syringe filters with 0.45 μm pore sizes
to remove insoluble components and then analyzed by an ion chromatography system (Dionex ICS
600, ThermoFisher Scientific, USA). Finally, the concentrations of three anions (Cl$^-$, SO$_4^{2-}$, and NO$_3^-$)
and five cations (Na$^+$, NH$_4^+$, K$^+$, Mg$^{2+}$, and Ca$^{2+}$) were obtained. The concentrations of OC and EC
in PM$_{2.5}$ samples were determined by analyzing a $1\times1.5$ cm$^2$ punch from each filter with an OCEC
analyzer (Model 5L, Sunset Laboratory Inc. USA), which adopted the NIOSH870 temperature
protocol with thermal–optical transmittance for charring correction. The OM concentration was
estimated via multiplying OC concentration by a factor of 1.6, based on the previous studies (Xing et
al., 2013; Zheng et al., 2015).
**2.3 AMS data analysis**

The HR-AMS V mode data were analyzed using standard data analysis software (PIKA V1.56D).

A constant collection efficiency (CE) of 0.5, similar to the previous studies conducted in winter at the
BJ site (Sun et al., 2014), was applied to the HR-AMS datasets to obtain the mass concentrations of
NR-PM$_1$ species. The relative ionization efficiencies used for OM, SO$_4^{2-}$, NO$_3^-$, NH$_4^+$, and Cl$^-$ were


1.4, 1.2, 1.1, 5.0, and 1.3, respectively. Positive matrix factorization (PMF) is a receptor model that
can identify potential sources without local source profiles provided (Xu et al., 2020). PMF was
performed on the high-resolution mass spectra of organics measured by HR-AMS. Six OA factors
were identified including fossil fuel-related OA (FFOA), cooking OA (COA), biomass burning OA
(BBOA), oxidized primary OA (OPOA), oxygenated OA (OOA), and aqueous-phase OOA (aqOOA).
Detailed information on the processing of HR-AMS data during the same campaign can be found in
the related paper (Xu et al., 2019).
**2.4 Individual particle analysis**

Individual particle samples were analyzed using TEM (JEM-2100, JEOL Ltd., Japan) operated

at a 200 kV accelerating voltage to acquire the morphology and size of individual particles and mixing
state (i.e., internally or externally mixed) of different aerosol components within one individual
particle. TEM is equipped with an energy-dispersive X-ray spectrometer (EDS, INCA X-Max$^N$ 80T,
Oxford Instruments, UK) to semi-quantitatively detect the elemental composition of individual
particles with atomic number greater than six ($Z \geq 6$). It should be noted that Cu peaks in the EDS
spectra are not considered due to the interference from the Cu substrate of TEM grids. The distribution
of aerosol particles on TEM grids is not uniform, with particle size decreasing from the center to the
edge of the distribution area. Therefore, to ensure the analyzed particles are representative, five grids
from the center to the edge of the particle distribution area in each sample were selected to conduct
TEM analysis. TEM images were manually processed by the RADIUS 2.0 software (EMSIS GmbH,
Germany) to determine the particle types, areas, perimeters, and equivalent circle diameters (ECD).
After a labor-intensive operation, a total of 1197 particles at the BJ urban site and 2443 particles at
the GC rural site were analyzed.

Scanning electron microscope (SEM, Ultra 55, Carl Zeiss Microscopy GmbH, Germany) was

operated at the 10 kV accelerating voltage and secondary electron (SE2) mode to observe the particle
surface topography. Furthermore, particles were imaged at a tilt angle of 75° to realize the



visualization of their morphology in the vertical dimension.

**2.5 Optical property calculation**

Mie theory has been widely used to calculate the optical properties of individual particles
(Chylek et al., 2019; Wu et al., 2018; Yu et al., 2019). In this study, the light absorption cross sections
(ACS) of the primary organic tar (POT, defined in Section 3.2) particles, as well as the coated POT
particles with secondary inorganic aerosol (SIA) shell (named as core–shell POT–SIA particle), at
the wavelength of 550 nm were calculated with BHCOAT Mie code (Bohren and Huffman, 1983).
Details for the classification of POT and POT–SIA particles, please refer to Section 3.2. For the core–
shell POT–SIA particles, a refractive index (RI) of $1.55-0i$ for non-light-absorbing SIA coating
(Denjean et al., 2014) and $1.67-0.27i$ for light-absorbing POT core (Alexander et al., 2008) were
adopted at the wavelength of 550 nm. The ECD of each POT–SIA particle and its POT core obtained
from the TEM images were used respectively as the input particle diameter ($D_p$) and core diameter
($D_c$) in the Mie model, which made the calculation sufficient to approximate reality. For the POT
particles (including bare POT particles and POT cores), the ECD of each POT particle and one-tenth
of it were input as the $D_p$ and $D_c$, respectively. Then by vanishing the refractive index difference
between the shell and core (i.e., POT shell and POT core, RI=$1.6-0.27i$), in which case we can obtain
the ACS of POT particles. In addition to the calculations for real individual particles observed from
the TEM images, we also constructed the model core–shell POT–SIA particle with different POT
core diameters (i.e., $D_c$=100, 200, 300, 400, 500, 800, 1300, and 1500 nm) and particle-to-core ratios
(i.e., $D_p/D_c$ ranged from 1 to 6 with an interval of 0.1), and calculated their ACS to explore the effects
of $D_c$ and $D_p/D_c$ changes on the light absorption enhancement factors ($E_{abs}$) of POT particles.

**3 Results and Discussion**

**3.1 Overview of a regional haze episode**

A typical regional heavy haze episode in the NCP was observed at the GC rural and BJ urban



sites during 22–27 November 2016. Fig. 2 shows time series of $PM_{2.5}$, aerosol chemical species, and
gaseous pollutants (i.e., CO, $SO_2$, $O_3$, and $NO_2$) at two sampling sites during the haze episode. Based
on the variations of hourly $PM_{2.5}$ concentrations, three pollution levels are defined: clean ($PM_{2.5} \leq 75$
$\mu g\ m^{-3}$), moderate pollution ($75\ \mu g\ m^{-3} < PM_{2.5} \leq 200\ \mu g\ m^{-3}$), and heavy pollution ($PM_{2.5} > 200\ \mu g$
$m^{-3}$). We classified clean period (21 Nov. 00:00 to 22 Nov. 19:00) and heavily polluted period (22
Nov. 20:00 to 27 Nov. 10:00) at the GC rural site; clean period (21 Nov. 00:00 to 24 Nov. 09:00),
moderately polluted period (24 Nov. 10:00 to 25 Nov. 16:00), and heavily polluted period (25 Nov.
17:00 to 27 Nov. 02:00) at the BJ urban site (Fig. 2). Furthermore, we divided the heavily polluted
period at the GC rural site into the early stage (22 Nov. 20:00 to 23 Nov. 20:00), middle stage (23
Nov. 20:00 to 24 Nov. 20:00), and late stage (24 Nov. 20:00 to 27 Nov. 08:00), based on the evolution
of chemical components in $PM_{2.5}$ (Fig. 2a). The averaged meteorological parameters and mass
concentrations of $PM_{2.5}$, aerosol chemical species, OA factors, and gaseous pollutants in different
periods at two sampling sites are summarized in Table S2.

The strong northwesterly winds ($> 4\ m\ s^{-1}$) accompanied with rain and snow invaded the NCP

during 20−21 November (Fig. S1), leading to the fast dispersion of air pollutants (Fig. 2). The low $T$
($-8$ to 5°C) and WS ($< 2\ m\ s^{-1}$) were displayed after the cold front (Fig. S1), which can facilitate the
accumulation of air pollutants (Zhong et al., 2019). At the GC rural site, $PM_{2.5}$ concentration began
to increase at 18:00 on 22 November and quickly reached a peak of 394 $\mu g\ m^{-3}$ within six hours (Fig.
2a). $PM_{2.5}$ chemical analysis reveals that OM (252.8 $\mu g\ m^{-3}$) accounted for 83% of the $PM_{2.5}$ at the
nighttime of 22 November (Fig. S2a), causing the fast transition from the clean to heavily polluted
period directly (Fig. 2a). In the early stage of heavily polluted period, the average $PM_{2.5}$ concentration
(288.3 $\mu g\ m^{-3}$) increased by a factor of seven compared with that (39.8 $\mu g\ m^{-3}$) in the clean period,
with OM being the largest contributor (185.1 $\mu g\ m^{-3}$) followed by SIA (i.e., sum of $SO_4^{2-}$, $NO_3^-$, and
$NH_4^+$; 36.4 $\mu g\ m^{-3}$) (Table S2). At the BJ urban site, the air quality remained clean before 24
November under continuous northerly winds (Figs. 2b and S1b). With prevailing winds changing





from northerly to southerly on 24 November (Fig. S1b), polluted air parcels in the south of NCP was
transported to Beijing (Fig. 1a), which has also been confirmed by another study conducted in the
same APHH-Beijing winter campaign (Du et al., 2019). Thus, the concentrations of $PM_{2.5}$, chemical
species in $NR-PM_1$, CO, and $SO_2$ at the BJ urban site increased simultaneously and sharply from
09:00 on 24 November, causing the transition from the clean period to the moderately polluted period
(Fig. 2b). The average $PM_{2.5}$ concentration in the moderately polluted period was 111.0 μg m$^{-3}$, 10
times higher than that (10.8 μg m$^{-3}$) in the clean period, and the OM and SIA contributed equally in
$NR-PM_1$ with their average concentrations being 44.4 and 43.4 μg m$^{-3}$, respectively (Table S2).
Following the haze evolution, the $PM_{2.5}$ levels increased gradually to 312.3 and 396.8 μg m$^{-3}$ in the
middle and late stages of heavily polluted period at the GC rural site and to 281.0 μg m$^{-3}$ in the
heavily polluted period at the BJ urban site (Fig. 2 and Table S2). Contrasting to the transition periods
at two sampling sites, we found that the SIA concentration increased significantly, meanwhile, the
OM concentration only slightly increased at the GC rural and BJ urban sites with the consistent
decreasing WS and increasing RH (Figs.2 and S2, Table S2). In a word, we observed that the SIA
fraction in fine particles increased and OM fraction decreased following the haze evolution (Fig. S2).

Figure S2a shows higher fractions of OM, EC, and Cl$^-$ at nighttime than daytime during the

whole haze episode at the GC rural site, suggesting the continuous strong local combustion emissions
at nighttime. Furthermore, the concentration of Cl$^-$ (8–22 μg m$^{-3}$) was much higher than that of K$^+$
(1–3 μg m$^{-3}$) (Fig. 2a), which suggests more contributions from coal combustion than biomass
burning at the GC rural site (Sun et al., 2014; Zhang et al., 2020). Based on the field investigation
and $PM_{2.5}$ analysis, we concluded that the explosive increase of $PM_{2.5}$ at the GC rural site was initiated
by the strong local emissions and accumulation of POA from residential coal combustion for heating
and a small fraction of biomass burning for cooking in rural areas. The PMF analysis shows that the
FFOA and BBOA (14.6–30.6 μg m$^{-3}$) contributed significantly (> 30%) to OM in the polluted period
at the BJ urban site (Fig. S3 and Table S2), suggesting that POA emitted in rural areas were



transported to Beijing under the southerly winds. In summary, the bulk analyses show that POA from
residential coal and biomass burning consistently contributed to the regional haze, and SIA produced
from the secondary formation had an increasing contribution at higher RH following the haze
evolution.
**3.2 Classification of individual particle types**
In this study, TEM observations show a distinct group of spherical and irregular primary organic
particles comprised of C, O, and Si elements during this haze episode (Fig. 3a). These particles are
stable under strong electron beams and appear as dark features in TEM images, which reflects their
high thickness and refractory properties (Ebert et al., 2016; Liu et al., 2018). The SEM image acquired
at a 75° tilt angle shows that these particles did not deform upon impaction and retained high vertical
dimensions (Fig. 4), indicating that these particles are in a solid state with high viscosity (Reid et al.,
2018; Wang et al., 2016). By contrast, the secondary particles (i.e., SIA and organic coating) became
flat on the substrate (Fig. 4). Previous studies only defined solid spherical tar balls emitted from coal
and biomass burning which are identified as light-absorbing POA (Adachi et al., 2019; C. Li et al.,
2019; Pósfai et al., 2003; Zhang et al., 2018). In this study, we observed abundant similar spherical
and some additional irregular POA particles sourced from coal and biomass burning in this haze
episode. To better represent the morphology and sources of these POA particles, we named them as
primary organic tar (POT) particles hereafter (Fig. 3a).
Besides, the typical individual particle types, such as SIA (Fig. 3b), mineral (Fig. 3c), soot (Fig.
3d), and fly ash/metal (Fig. 3e) particles were also classified during this haze episode. The detailed
classification criteria of these particle types derived from the TEM images and their sources can be
found in our previous paper (Li et al., 2016a). It should be noted that some SIA particles were coated
with secondary organic coatings (Fig. 3b) which were produced from the chemical oxidation of
volatile organic compounds (Li et al., 2016b). TEM observations further show the internal mixture of
POT or soot particles with SIA, i.e., POT–SIA (Fig. 3f) and soot–SIA (Fig. 3g). To better understand



the number variations of different particle types, we further classified the POT and POT–SIA particles
as the POT-containing particles, and soot and soot–SIA particles as soot-containing particles.
**3.3 Relative abundance of individual particle types**
Figure 5 shows number fractions of different particle types in different periods at GC rural and
BJ urban sites. At the GC rural site, POT-containing and soot-containing particles were the major
particle types with their corresponding contributions being 37.6% and 35.9% by number, followed
by SIA particles (22.4%) in the clean period. When the haze episode occurred at the GC rural site,
POT-containing particles became dominant in the early stage of heavily polluted period and its
number fraction (64.8%) was nearly twice that (37.6%) in the clean period (Fig. 5a). This result
consists well with the bulk PM$_{2.5}$ analysis that shows a sharp increase in OM concentration in the
early stage of heavily polluted period (Fig. 2a). With the increasing pollution level from the early
stage to the late stage of heavily polluted period, the fraction of POT-containing particles slightly
decreased from 64.8% to 50.8%, by contrast, the fraction of SIA particles increased from 4.6% to
12.4% (Fig. 5a). The variations of POT-containing and SIA particles are similar to the results from
the bulk PM$_{2.5}$ analysis shown in Fig. 2a.
At the BJ urban site, the contribution of POT-containing particles (15.1%) in the clean period
was much lower than that (37.6%) at the GC rural site (Fig. 5). Following the transition from the
clean period to the moderately polluted period at the BJ urban site, the fraction of POT-containing
particles (66.2%) increased significantly by more than a factor of four compared with that (15.1%) in
the clean period. Meanwhile, the fractions of soot-containing, mineral, and SIA particles decreased
largely. When the pollution level changed to the heavily polluted period, similar to the situation at the
GC rural site, the fraction of SIA particles increased from 7.8% to 13.2% and the fraction of POT-
containing particles decreased slightly from 66.2% to 52.8% (Fig. 5b). Overall, the results from the
individual particle analysis consist well with the changes in aerosol chemical components obtained
by the bulk analysis as shown in Fig. 2. Furthermore, individual particle analysis reveals that the POT-



containing particles dominated (> 50% by number) in the rural and urban air during the regional
wintertime haze episode.

**3.4 Atmospheric aging of POT particles following the haze evolution**

The TEM images clearly show the morphology and mixing state of individual particles in
different polluted periods at GC rural and BJ urban sites (Fig. 6). At the GC rural site, TEM
observations reveal abundant bare POT particles in the early stage of heavily polluted period (Fig.
6a). These POT particles have been proved to be emitted from the residential coal and biomass
burning in the wintertime of northern China in our previous studies (Chen et al., 2017; Zhang et al.,
2018). Based on the integrated analyses of individual particles and bulk samples, we confirmed that
large amounts of POT particles emitted from the intense domestic coal and biomass burning for
heating and cooking significantly contributed to the deterioration of the air quality in rural areas.
When the haze episode evolved into the late stage of heavily polluted period, we found that most of
the POT particles were coated with SIA (i.e., POT–SIA particle) forming the core–shell structure (Fig.
6b). Large amounts of POT particles in the regional haze layer provided surfaces for the
heterogeneous reactions of $SO_2$ and $NO_x$, which promotes the formation of SIA on POT particles in
the humid polluted air (Ebert et al., 2016; Zhang et al., 2017).
Following the regional transport of polluted air masses from the south to the north of the NCP,
abundant POT particles also occurred in the moderately polluted period at the BJ urban site (Fig. 6c).
Therefore, we conclude that the POT particles emitted in the rural areas in the south of the NCP could
be transported to the BJ urban site and significantly affect the air quality. Following the haze evolution,
similar to those at the GC rural site, the POT particles aged and became core–shell POT–SIA particles
at the BJ urban site in the heavily polluted period (Fig. 6d).
Based on the mixing state of POT-containing particles, we found that following evolution of the
haze episode, the fraction of bare POT particles was reduced by twice from 91.4% in the early stage
to 39.6% in the late stage of heavily polluted period at the GC rural site, and the fraction of POT–SIA



particles correspondingly increased by seven times from 8.6% to 60.4% (pie charts in Fig. 7).
Similarly, at the BJ urban site, the fraction of bare POT particles decreased from 70.4% in the
moderately polluted period to 31.4% in the heavily polluted period, and the fraction of POT–SIA
particles increased correspondingly from 29.6% to 68.6% (pie charts in Fig. 7). Consequently, the
average size of POT-containing particles changed from 505 nm in the early stage to 837 nm in the
late stage of heavily polluted period at the GC rural site and from 443 nm in the moderately polluted
period to 732 nm in the heavily polluted period at the BJ urban site (Fig. 7a). Interestingly, the average
sizes of POT particles (including POT cores and bare POT particles) remained similar following the
haze evolution, with respective values being 469, 508, and 465 nm in the early, middle, and late stages
of heavily polluted period at the GC rural site and 381 and 379 nm in the moderately and heavily
polluted periods at the BJ urban site (Fig. 7b). The average sizes of POT particles at the BJ urban site
were slightly smaller than those at the GC rural site, which is reasonable because the POT particles
collected at the GC rural site were close to the emission sources and larger particles are more likely
to be removed during the regional transport (Seinfeld and Pandis, 2006). Adachi et al. (2018) reported
that tar balls retained their spherical shapes and the particle masses and sizes did not change largely
when heated to 300°C in TEM. As a result, we conclude that the POT particles should be quite
physically stable and chemically inert in the atmosphere, which can be transported long distances.

The $D_p/D_c$ ratio can be used to indicate the aging degree of POT-containing particles in the

atmosphere (Chen et al., 2017; Li et al., 2011). By calculating the $D_p/D_c$ ratio, we realized the
quantification of the aging degree of POT-containing particles as shown in Fig. 8. In the early stage
of heavily polluted period at the GC rural site, the POT-containing particles were freshly emitted bare
POT particles with a fraction of 91.4% (Fig. 7), therefore, the average $D_p/D_c$ ratio was close to one
(1.02). Following the haze evolution at the GC rural and BJ urban sites, the average $D_p/D_c$ ratios
increased from 1.08 in the middle stage to 1.60 in the late stage of heavily polluted period at the GC
rural site, and from 1.11 in the moderately polluted period to 1.67 in the heavily polluted period at





the BJ urban site. The results indicate that the POT particles were thickly coated with SIA due to the
particle aging process in the haze. Here we can obtain two conclusions based on the individual particle
analysis: (1) more POT particles continuously aged and were coated with SIA following the haze
evolution; (2) the SIA coating gradually grew through the heterogeneous conversion of gaseous
precursors (e.g., $SO_2$ and $NO_x$) in the polluted air. Therefore, aging processes of individual POT
particles in wintertime haze well reflect the regional haze evolution in the NCP.
**3.5 Changes in light absorption of the POT particles**
It is well known that organic aerosols emitted from the coal and biomass burning are the main
source of light-absorbing BrC (M. Li et al., 2019; Lin et al., 2016; Sun et al., 2017). Recently, some
observation and modeling works show that the BrC in haze layers over the NCP can affect the regional
energy budget (Feng et al., 2013; Wang et al., 2018; Xie et al., 2019). However, there is no answer on
how the aging process of POT particles influence their optical absorption in the regional haze. Here
we further explored variations in the optical absorption of individual POT particles using Mie theory
following the haze evolution at the GC rural and BJ urban sites (Fig. 9).
At the GC rural site, the average ACS of individual POT-containing particles ($ACS_{POT\text{-}containing}$)
under the actual scenario in the early, middle, and late stages of heavily polluted period were estimated
to be $3.09\times10^{-14}$, $3.97\times10^{-14}$, and $4.43\times10^{-14}$ $m^2$, respectively (Fig. 9a). If all the POT-containing
particles were not coated with SIA in each period (i.e., no particle aging scenario), the corresponding
average ACS of individual POT particles ($ACS_{POT}$) were $3.01\times10^{-14}$, $3.52\times10^{-14}$, and $3.18\times10^{-14}$ $m^2$,
respectively (Fig. 9a). Based on the ratios of $ACS_{POT\text{-}containing}$ to $ACS_{POT}$, we obtained that the $E_{abs}$
were 1.02, 1.12, and 1.39 in the early, middle, and late stages of heavily polluted period, respectively,
at the GC rural site (Fig. 9a). Similarly, at the BJ urban site, the $E_{abs}$ were 1.10 and 1.39 in the
moderately and heavily polluted periods, respectively, with the corresponding average $ACS_{POT\text{-}}$
$_{containing}$ being $2.06\times10^{-14}$ and $3.00\times10^{-14}$ $m^2$ and $ACS_{POT}$ being $1.86\times10^{-14}$ and $2.15\times10^{-14}$ $m^2$ (Fig.
9b). It should be noted that the light absorption capacity of individual POT particles was a little lower





at the BJ urban site than that at the GC rural site (Fig. 9), which was mainly attributed to the smaller
sizes of POT particles at the BJ urban site (Fig. 7).

To better understand the influence of SIA-coating thickness and POT-core diameter on the light

absorption of POT–SIA particles, we modeled the variations in $E_{abs}$ of POT–SIA particles (i.e., ratios
of ACS$_{POT–SIA}$ to ACS$_{POT\ pore}$) with different $D_c$ as a function of $D_p/D_c$ ratios (Fig. 10). The results
show that $E_{abs}$ is sensitive to the changes in both $D_c$ and $D_p/D_c$ ratio. When $D_p/D_c$ < 1.5, the $E_{abs}$
increases sharply with the increase of $D_p/D_c$ ratio for different POT core sizes; but when $D_p/D_c$ > 1.5,
the $E_{abs}$ does not show an increase any more for particles with $D_c$ > 200 nm, and the $E_{abs}$ is limited to
between 1.5 and 2 for particles with $D_c$ ranging from 200 to 1500 nm (Fig. 10). The diameters of the
observed POT cores at GC rural and BJ urban sites in this study were mainly in the range of 200 to
800 nm (Fig. 7), thus the $E_{abs}$ of the observed POT–SIA particles in the NCP were mostly below 1.75
(Fig. 10). All the above results indicate that the atmospheric aging process could significantly improve
the light absorption capacity of POT particles along with the evolution of haze episodes due to the
"lensing effect" of SIA coating.
**4 Conclusions and implications**

This study demonstrates that the primary pollutants especially large amounts of POT particles

emitted from the residential coal and biomass burning in rural areas initiated the wintertime regional
haze episode in the NCP. The presence of abundant POT particles in the atmosphere could further
provide surfaces for the heterogeneous reactions promoting the large production of SIA under
stagnant metrological conditions with high RH, which further elevated the pollution level. Compared
with the tar balls which have been confirmed as BrC with strong light-absorbing capacities in previous
studies (Adachi et al., 2019; C. Li et al., 2019; Zhang et al., 2018), the spherical or irregular POT
particles observed in this study can better represent various light-absorbing primary organic particles
in the wintertime hazes. Therefore, the ubiquitous brown POT particles in the atmosphere of NCP
unquestionably affect the energy balance (Feng et al., 2013). We found that the POT particles



remained quite stable during the regional transport from the rural areas to urban Beijing in the NCP
and were coated with SIA through the atmospheric aging process in the haze layer, which could
significantly enhance the light absorption capacity of POT particles via the "lensing effect" of SIA
coating. We estimated that the $E_{abs}$ values were within the upper limit of 1.75 in core–shell Mie
simulations considering the typical size distribution (200–800 nm) of POT particles in the NCP.
Furthermore, Alexander et al. (2008) found plenty of primary brown organic particles with strong
light absorption capacity in East Asian outflow, which indicates that the POT particles could be
transported over long distances and still retain their strong light-absorbing properties, and thus can
affect the regional and even global radiative forcing. Therefore, we highlight that the "lensing effect",
which has been adequately reported on BC particles but not on the POT particles in previous studies,
should be further considered on the POT particles in radiative forcing models.
The primary pollutants from the intense coal and biomass burning in rural areas can also pose
serious threats to human health. In particular, large amounts of toxic primary particles can be released
from coal and biomass burning, such as the polycyclic aromatic hydrocarbons (PAHs) in POA and
the toxic heavy metals (Cheng et al., 2019; C. Li et al., 2019; X. Wang et al., 2017), which could lead
to high concentrations of toxic substances in the rural atmosphere and further be transported in large
scale. Recently, Zhao et al. (2018) reported that approximately 80% of premature deaths occurred in
the rural areas of China in 2015 was attributed to the $PM_{2.5}$ released from household fuels.
Considering the adverse effects of residential coal and biomass burning on the haze formation,
human health, and climate change, we suggest that the governments should continue to implement
the "Clean Air Actions" (Zhang and Geng, 2019), especially encourage the use of clean energy such
as electricity and natural gas for heating and cooking in rural areas of North China in winter.
**Data availability**
All data presented in this paper are available upon request. Please contact the corresponding
author (liweijun@zju.edu.cn).



## Author Contributions

WL and LL designed the research. LL performed the data analysis and wrote the manuscript and WL revised it. JZ and YZ assisted with the sample collection. YS provided the AMS data at the Beijing site. LL, JZ, YZ, LX, QY, and YW carried out the chemical analysis of $PM_{2.5}$ and TEM analysis of individual particles. ZS, YS, DL, and PF contributed to the improvement of this manuscript. All the authors approved the final version of this paper.

## Competing interests

The authors declare that they have no conflict of interest.

## Acknowledgements

This work was funded by the National Key R&D Program of China (2017YFC0212700), National Natural Science Foundation of China (42075096, 91844301), and Zhejiang Provincial Natural Science Foundation of China (LZ19D050001). Z. Shi acknowledges the UK Natural Environment Research Council (NE/S00579X/1 and NE/N007190/1). We acknowledge the NOAA Air Resources Laboratory for the provision of the HYSPLIT transport and dispersion model and READY website (http://www.ready.noaa.gov) used in this paper.

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

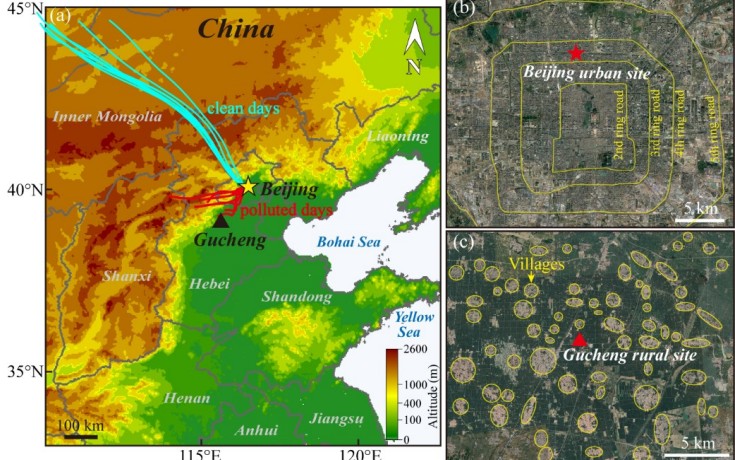


Figure 1. Locations of Beijing and Gucheng in the North China Plain (a) and the expanded view of surrounding

topographies around the Beijing urban site (b) and Gucheng rural site (c). The 24-h backward trajectories of air

masses ending at the height of 100 m (a.g.l) over the Beijing urban site in clean and polluted days during 20–27

November, 2020 are also shown in (a). (Map copyright @2020 Google Maps)



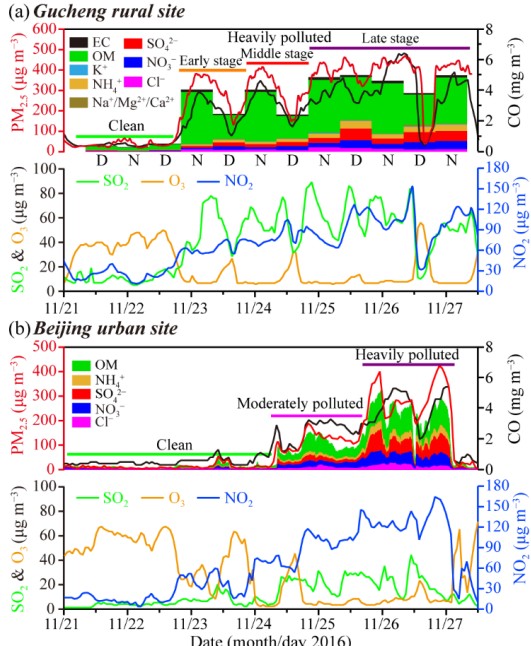

Figure 2. Time series of PM$_{2.5}$, aerosol chemical species, and gaseous pollutants (i.e., CO, SO$_2$, O$_3$, and NO$_2$) at the

(a) Gucheng rural site and (b) Beijing urban site. Chemical species at the rural site were obtained by offline analysis

of daytime (D) and nighttime (N) PM$_{2.5}$ filter samples. Chemical species at the urban site were obtained by online

analysis of NR-PM$_1$ using a high-resolution aerosol mass spectrometer (HR-AMS). The different periods of the

haze episode at rural and urban sites are marked in this figure.



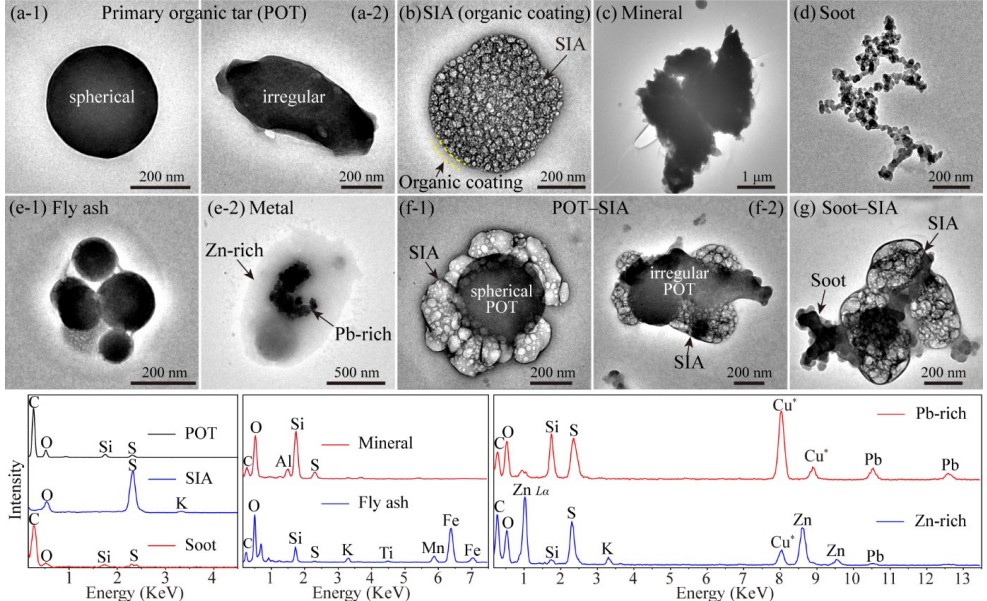

Figure 3. Typical transmission electron microscopy (TEM) images and energy-dispersive X-ray spectrometry (EDS) spectra showing the morphology, composition, and mixing structures of different individual particle types. (a) primary organic tar (POT) particles with (a-1) spherical and (a-2) irregular shapes; (b) secondary inorganic aerosol (SIA) particle with secondary organic coating; (c) mineral; (d) soot; (e-1) fly ash and (e-2) metal; (f) internally mixed POT particle with SIA coating (POT–SIA); (g) internally mixed soot particle with SIA coating (soot–SIA).



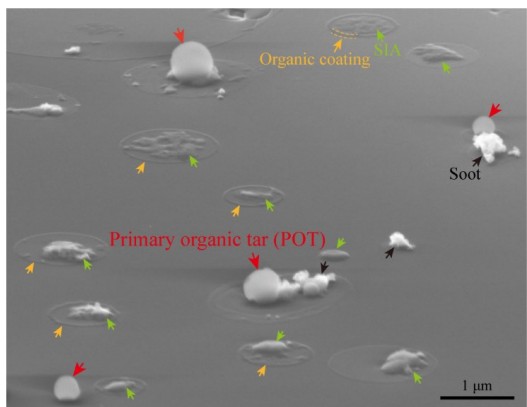

Figure 4. Scanning electron microscopy (SEM) image acquired in the secondary electron (SE2) mode at a 75° tilt

angle showing the surface morphology of individual particles in the vertical dimension. The red, black, green, and

orange arrows indicate the primary organic tar (POT) particle, soot particle, secondary inorganic aerosol (SIA)

particle, and secondary organic coating, respectively.





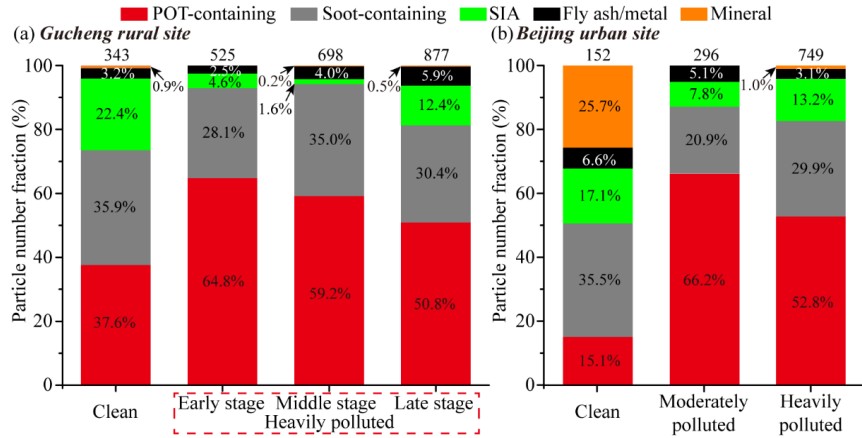

Figure 5. Relative abundance of different particle types in different periods at the (a) Gucheng rural site and (b)

Beijing urban site. The numbers of analyzed particles in different periods are shown on the top of each column.



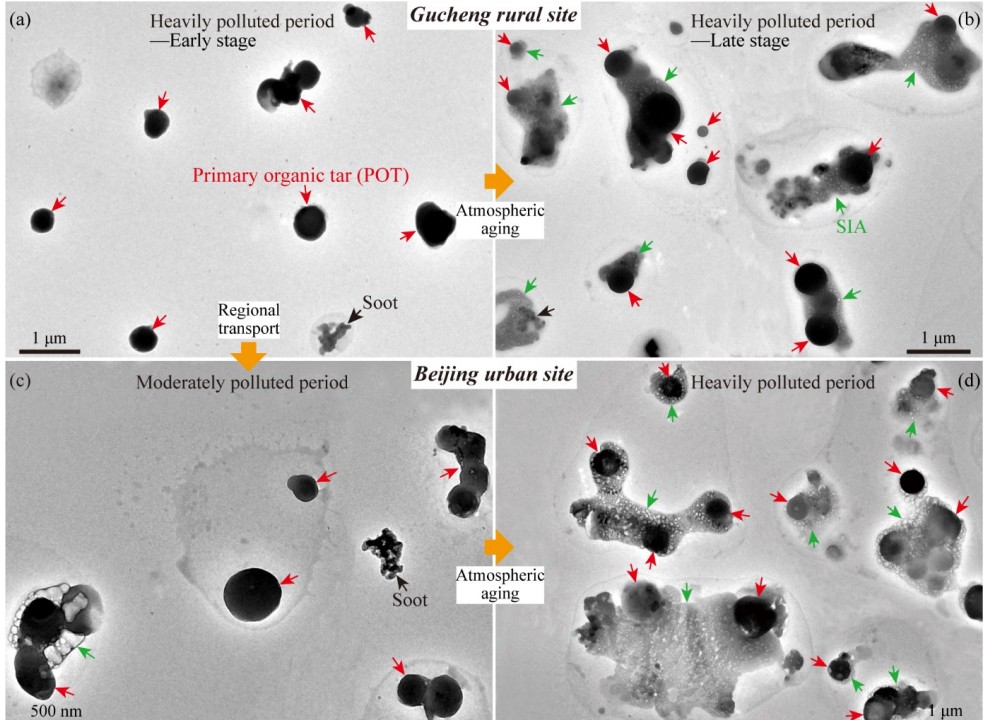

712

Figure 6. TEM images showing individual particles collected in the (a) early stage and (b) late stage of heavily

polluted period at the Gucheng rural site and in the (c) moderately polluted and (d) heavily polluted periods at the

Beijing urban site. The red, green, and black arrows indicate the primary organic tar (POT) particle, secondary

inorganic aerosol (SIA) particle, and soot particle, respectively.



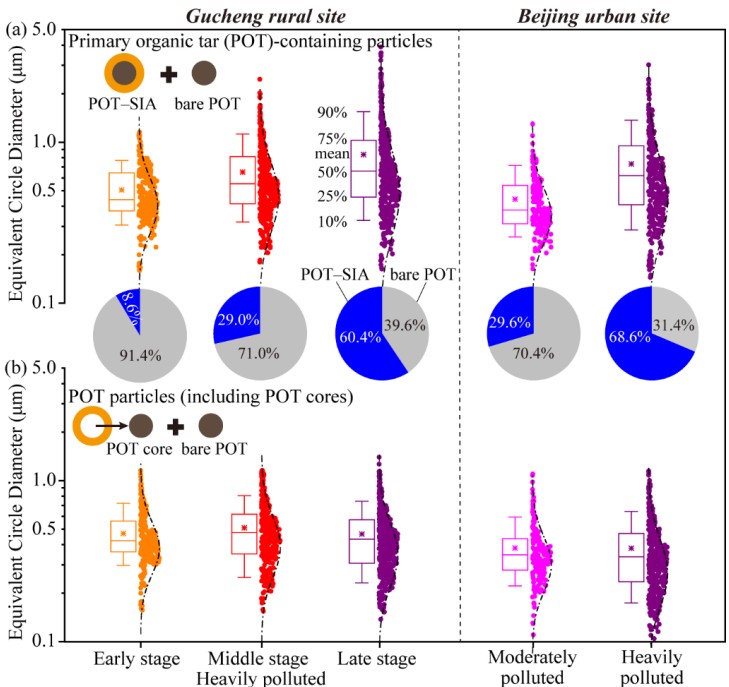

Figure 7. Box plots of equivalent circle diameters (ECD) of (a) primary organic tar (POT)-containing particles (i.e.,
coated POT with secondary inorganic aerosol shell (named as POT–SIA) and bare POT) and (b) POT particles
(including POT cores and bare POT) in different polluted periods at the Gucheng rural site and Beijing urban site.
The solid circles (right of the box) represent the ECD of individual particles with lognormal distributions. The pie
charts present the variation in relative number fractions between POT–SIA and bare POT in different polluted
periods.

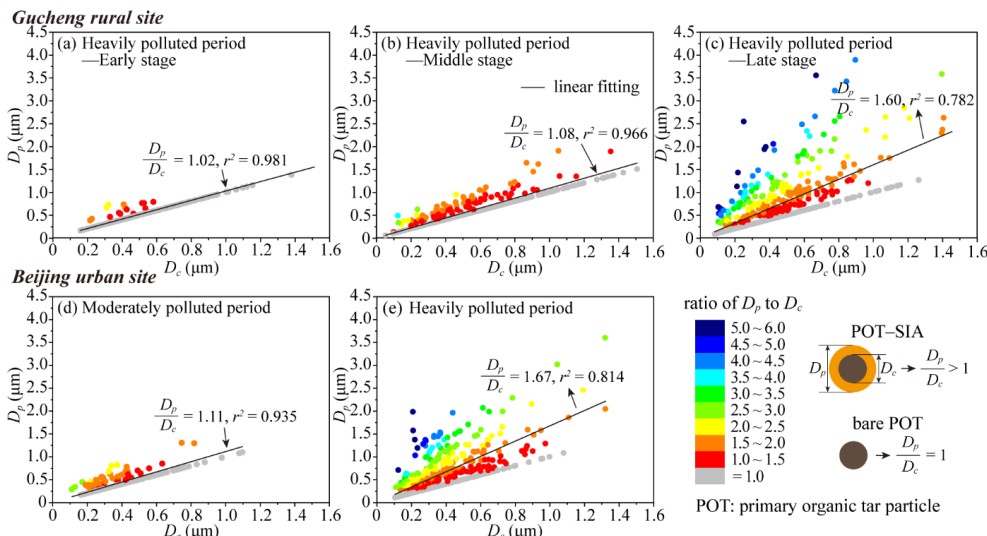

Figure 8. Relationship between the diameter of POT-containing particle ($D_p$) and its POT core ($D_c$) in the (a) early stage, (b) middle stage, and (c) late stage of heavily polluted period at the Gucheng rural site and in the (d) moderately polluted and (e) heavily polluted periods at the Beijing urban site.

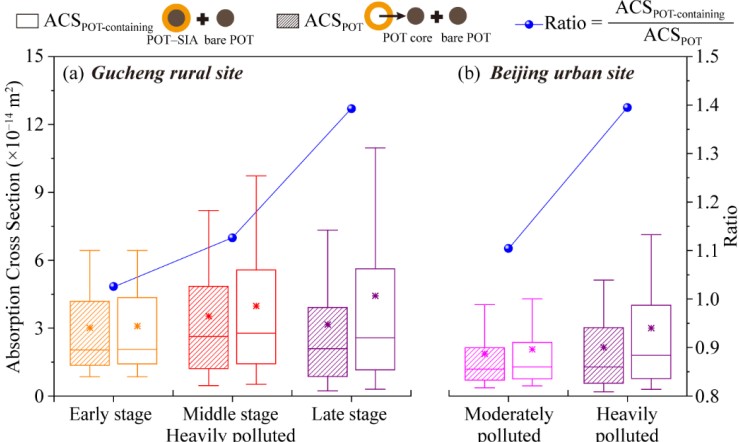


Figure 9. Box plots of absorption cross sections (ACS) of individual POT-containing particles (i.e., ACS_POT-containing

under the actual scenario) and POT particles (including POT cores, i.e., ACS_POT under the no particle aging scenario),
and the variation in the ratios of ACS_POT-containing to ACS_POT in different polluted periods at the (a) Gucheng rural site
and (b) Beijing urban site. The box represents the 25th (lower line), 50th (middle line), and 75th (top line) percentiles;
the asterisk in the box represents the mean value; the end lines of the vertical bars represent the 10th (below the
box) and 90th (above the box) percentiles.



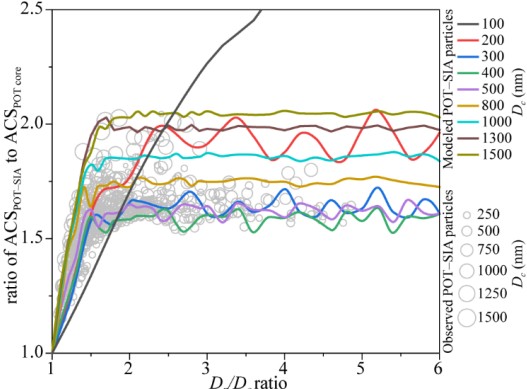


Figure 10. Mie-simulated light absorption enhancements ($E_{abs}$) of modeled POT–SIA particles (i.e., ratio of ACS$_{POT–}$

$_{SIA}$ to ACS$_{POT\ pore}$) with different POT core diameters ($D_c$) as a function of particle-to-core ratio ($D_p/D_c$ ratio) at the

wavelength of 550 nm (solid lines). The open circles represent all the POT–SIA particles observed during the whole

polluted periods at Gucheng rural and Beijing urban sites.