# Peer review of "Persistent residential burning-related primary organic particles during"

_Atmospheric Chemistry and Physics, 2020_

## Referee Comment (RC1) · Anonymous Referee #1 · 12 Nov 2020

In this study, an aging and optical change of organic tar particles in the regional haze was observed. Domestic coal and biomass burning are suggested to be the important reasons for haze formation in the NCP. It is found that with the evolution of haze, organic particles decreased, secondary inorganic aerosol increased, POT-SIA particles increased, and POT (primary organic tar) particles decreased, indicating that POT particles could provide surface for heterogeneous reactions of $SO_2$ and $NO_x$. It is also concluded that POT particles are coated with secondary inorganic aerosol, which leads to increased light absorption of particulate. Therefore, the "lensing effect" should be further considered on the POT particles in radiative forcing models. The results obtained in this study are interesting, and worthy to be published in ACP.

[Figure]

Followings are my suggestions to the manuscript: 1. The diagrams are too complex to understand, and some of the parameters in the diagrams have not been analyzed. For example, the mass concentrations of CO and NO2 in Figure 2. 2. The offline bulk sample analysis is used for rural site while the online analysis is used for urban site. Can the values from two methods for the two sites be compared? Are there any errors or deviations between the results from offline bulk sample analysis and online analysis for urban site? 3. The analysis shows that fossil fuel burning and biomass burning are the main sources of pollution. What is the difference between POT particles produced by these two sources? 4. It is concluded that the particle size of the rural point is larger than that of the urban point, because the particles with small particle size are easier to be transmitted. However, in this transportation process, the formation of new particles or secondary chemical reaction (aging) may occur, which would increase the particle sizes, do the authors take into account this factorïij§ 5. The second paragraph of the conclusion: "The primary pollutants from the intense coal and biomass burning in rural areas can also pose serious threats to human health . . . . . .". This looks unnecessary as the health effects are not the focus of this current study and it may need to be removed. 6. Line251ïijŇ "Figure S2a shows higher fractions of OM, EC, and Cl− at nighttime than daytime during the whole haze episode at the GC rural site, suggesting the continuous strong local combustion emissions at nighttime". We notice that the whole haze period was from November 22 to 27, and Figure 2A shows that the EC quality score was higher at nighttime than in the daytime on November 22. Please also explain why the nighttime would have hither level combustion emissions. 7. Figure S2b shows only the ion concentration of 5 parameters, and the concentration of other ions at the rural points should also be listed. 8. It is better to have discrimination of the ion concentration between daytime and nighttime for Figure S2 and Figure 2. In addition, the variation trend of CO concentration in Figure 2 may needs to be put together with gaseous pollutants such as SO2 and O3. 9. The specific calculation formula of light absorption cross section needs to be provided in methodology section. 10. Figure 9. Legend error. Ratio should be replaced by Eabs. 11. Line 390: "pore" may be core

---

## Referee Comment (RC2) · Anonymous Referee #2 · 18 Nov 2020

This study describes and compares wintertime aerosol particles in a rural location on the North China Plane with aerosol in Beijing (downwind from the rural site): aerosol concentration, chemical composition and individual-particle properties were measured, and optical properties simulated. The measurement campaign covered a period when conditions changed from relatively clean to heavily polluted, and the aerosol at the assumed source (the rural site) and in the urban haze could be compared. The major finding is that refractory primary organic particles from coal and biomass burning dominate at the rural site, and similar but coated particles are the major constituents in the aged haze. The inorganic coatings change the optical properties of the aerosol, by enhancing their absorption. The paper is sound and fairly well written. I have three

concerns, one of which is partly a terminological issue.

(1) Primary Organic Tar (POT). I am slightly uncomfortable with the use of the term "primary organic tar" (and its acronym, POT) for the observed class of particles. From the results it seems a correct interpretation that these particles are primary organics emitted by burning of coal and biomass. However, this is an interpretation and not an observation, as is now presented right at the start of the paper. The properties we know from TEM include the morphologies of the particles (some spherical and others irregular), their C-rich nature (although the particles were collected on a thick carbon+plastic film and therefore we do not know how much C they contain), and their thickness from SEM which suggests they were quite viscous when they arrived onto the grids. Indeed, these properties and the high concentrations at the locality where burning of coal is a major source of particles suggests they are primary organics. However, I am not sure they can be called "tar", since there is no study of the speciation of their components (although the authors refer to Xu et al. 2019 who used AMS and identified three groups of burning-related factors). Admittedly, "tar" is a vaguely defined substance. However, the atmospheric chemistry literature is already replete with vaguely defined terms (and acronyms), and introducing a new one, such as POT may not help to clear the confusion. For the spherical types of such particles already exists a widely used term, "tar ball". And about tar balls we know they are primary (Tóth et al. ACP 14, 6669–6675, 2014), indeed made of tar (Tóth et al. ACP 18, 10407–10418, 2018), and even their optical properties are known (Hoffer et al. ACP 16, 239–246, 2016). By using POT, I feel the definition of tar ball is diluted and thereby the utility of the term diminished. In summary, I believe it may be more prudent to call the POT particles just burning-related primary organics (PO), and it could be mentioned that the spherical ones appear to be tar balls. But, in the end, it is up to the authors what they wish to call these particles.

(2) Optical properties. The calculations for coated absorbing organic particles are interesting but I do not completely understand Figure 10. (1) What is the explanation for the insensitivity of ACS enhancement to the Dp/Dc size ratios larger than about 1.5?

(2) What is the reason for the apparently random variation of ACS enhancement with Dc? I wonder if any optical measurements were done under hazy conditions during the sampling campaign? If such results are available, it would be interesting to compare them with the models.

(3) Size changes. If the particles were transported from the rural site to Beijing, and during transport secondary aerosol formed on them, it does not see logical that their sizes decreased. The explanation offered is that large particles deposited but this process does not seem feasible for the particles in this size range. Can you please provide some more discussion on this issue?

Minor comments:

29: "average particle-to-core ratios.."- it is unclear what ratios; probably "..diameter ratios" would be correct

32: replace "we using.." with "Using Mie theory we estimated.. "

61: chromatography

64-66. subject of sentence unclear

85 "properties" instead of "information"

178-179 "five grids.. were selected" - I guess "five grid meshes. . . were selected" would be correct

191: primary organic tar, POT: I do not understand the utility of this name and abbreviation. First, tar should be organic anyway, so "organic tar" is a tautology. Second, how do you know it is tar?

195-197: Hoffer et al. ACP 16, 239–246, 2016 found 1.84 - 0.21i for tar balls - I wonder if this would make a large difference in your calculations, compared to the value by Alexander et al.

246-249: unclear sentence, please reword

266-267 and text that follows: "TEM observations show a distinct group of spherical and irregular primary organic particles.." - I think TEM observations indeed show the morphologies of the particles but they do not provide direct information on their "organic" nature (only their elemental composition), let alone their primary origin. Please try to distinguish observations from interpretation.

278: Could it be just PO?

295: "agrees" instead of "consists"

372: influences

376-377: how were the absorption cross sections estimated?

405-409: this seems important but I do not understand the statement completely.

---

## Author Comment (AC1) · 1 Jan 2021

**Reply to Referee#1:**

General Comments:

In this study, an aging and optical change of organic tar particles in the regional haze was observed. Domestic coal and biomass burning are suggested to be the important reasons for haze formation in the NCP. It is found that with the evolution of haze, organic particles decreased, secondary inorganic aerosol increased, POT-SIA particles increased, and POT (primary organic tar) particles decreased, indicating that POT particles could provide surface for heterogeneous reactions of  $SO_2$  and  $NO_x$ . It is also concluded that POT particles are coated with secondary inorganic aerosol, which leads to increased light absorption of particulate. Therefore, the "lensing effect" should be further considered on the POT particles in radiative forcing models. The results obtained in this study are interesting and worthy to be published in ACP.

Response: We appreciated referee#1's careful reading and positive feedback. All the comments and suggestions are valuable for improving the quality of our paper. Our point-to-point replies to the referee's comments are listed below and the changes are incorporated into the revised manuscript marked in red color.

Specific Comments:

 The diagrams are too complex to understand, and some of the parameters in the diagrams have not been analyzed. For example, the mass concentrations of CO and NO2 in Figure 2.

Response: According to the referee's suggestion, we redraw the figures. Because the variations of the gaseous pollutants were used in this manuscript only to help understand the evolution of haze episodes, they were not the main concerns of this manuscript, therefore we did not discuss them in detail and we moved the lines of gaseous pollutants in Figure 2 into the Supplementary Information (Figure S2) to make the figure more concise to read.

2. The offline bulk sample analysis is used for rural site while the online analysis is used for urban site. Can the values from two methods for the two sites be compared? Are there any errors or deviations between the results from offline bulk sample analysis and online analysis for urban site?

Response: Thanks for the comment. In this paper, we mainly focused on the microscopic properties and aging process of primary organic particles through the individual particle analysis. Due to the limited instruments, we collected offline PM2.5 filter samples at the Gucheng rural site and used the Aerosol Mass Spectrometer (AMS) to continuously monitor the chemical species at the Beijing urban site. The bulk analysis results were used as solid evidence to support the individual particle analysis results. We mainly focused on the relative changes of different chemical species but not the absolute concentration changes. The detailed comparison of bulk analysis results between the two sites is not intended. Therefore, we did not compare the bulk analysis results between the two sites in this paper. The deviations between the offline bulk sample analysis and the online analysis at the urban site were not determined because of the lack of offline samples.

3. The analysis shows that fossil fuel burning and biomass burning are the main sources of pollution. What is the difference between POT particles produced by these two sources?

Response: The previous papers reported that tar balls can be emitted from biomass burning (Pósfai et al., *J. Geophys. Res.*, 108, D13, 8483, 2003) and coal combustion (Zhang et al., *J. Geophys. Res.*, 123, 12964-12979, 2018). These papers reported similar characteristics such as spherical morphology, high viscosity, and C/O composition of tar balls. Therefore, it is difficult to distinguish the tar balls emitted from biomass burning and coal combustion, which is out of the main objective of this manuscript. Therefore, we did not identify their exact sources, and as referee#2 suggested, we refer to them as the burning-related primary organic particles. We think that the reviewer's comments might be our next aim to work on it. References:

Pósfai, M., Simonics, R., Li, J., Hobbs, P. V., and Buseck, P. R.: Individual aerosol particles from biomass burning in southern Africa: 1. Compositions and size distributions of carbonaceous particles, J. Geophys. Res.-Atmos., 108, 8483, 10.1029/2002JD002291, 2003.

Zhang, Y., Yuan, Q., Huang, D., Kong, S., Zhang, J., Wang, X., Lu, C., Shi, Z., Zhang, X., Sun, Y., Wang, Z., Shao, L., Zhu, J., and Li, W.: Direct observations of fine primary particles from residential coal burning: insights into their morphology, composition, and hygroscopicity, J. Geophys. Res.-Atmos., 123, 12964-12979, 10.1029/2018JD028988, 2018.

4. It is concluded that the particle size of the rural point is larger than that of the urban point, because the particles with small particle size are easier to be transmitted. However, in this transportation process, the formation of new particles or secondary chemical reaction (aging) may occur, which would increase the particle sizes, do the authors take into account this factor.

Response: We appreciate the referee's comment. Indeed, the whole aged particle sizes increased due to the production of secondary aerosols on their surfaces during the particle aging process as we have shown in the manuscript. What we refer to is that the sizes of uncoated POA particles slightly decreased during the regional transport.

5. The second paragraph of the conclusion: "The primary pollutants from the intense coal and biomass burning in rural areas can also pose serious threats to human

health.....". This looks unnecessary as the health effects are not the focus of this current study and it may need to be removed.

Response: According to the referee's suggestion, we removed this part.

6. Line 251 "Figure S2a shows higher fractions of OM, EC, and Cl- at nighttime than daytime during the whole haze episode at the GC rural site, suggesting the continuous strong local combustion emissions at nighttime". We notice that the whole haze period was from November 22 to 27, and Figure 2A shows that the EC quality score was higher at nighttime than in the daytime on November 22. Please also explain why the nighttime would have higher level combustion emissions.

Response: As we have written in the manuscript, at the GC rural site the  $PM_{2.5}$  concentration began to increase at 18:00 on 22 November and the air quality was changed from the clean period to heavily polluted period during the 22–27 November. The EC concentration at the nighttime (12.1 µg m-3) was much higher than that (1.8 µg m-3) in the daytime on November 22 (Figure 2a). In rural areas, people may work out and the heating activities stopped in the daytime. But at nighttime people all stayed at home and consumed more biomass and coal for cooking and heating, especially the heating activities with coal maintained several hours during the nighttime, which can release large amounts of pollutants into the air. Therefore, the primary emissions were much higher during the daytime than the nighttime.

7. Figure S2b shows only the ion concentration of 5 parameters, and the concentration of other ions at the rural points should also be listed.

Response: As we have mentioned above, at the Gucheng rural site, offline  $PM_{2.5}$  filter samples were collected to obtain the concentrations of water-soluble inorganic ions (i.e., Cl-, SO42-, NO3-, Na+, NH4+, K+, Mg2+, and Ca2+) using the ion chromatograph and obtain the OC and EC concentrations using the OC/EC analyzer. However, at the Beijing urban site offline  $PM_{2.5}$  filter samples were not collected. Instead, we used the online Aerosol Mass Spectrometer (AMS) which can only obtain five chemical species (i.e., OM, Cl-, SO42-, NO3-, and NH4+). Therefore, we provided 10 chemical species at the Gucheng rural site and five chemical species at the Beijing urban site. We should note that the lack of Na+, Mg2+, Ca2+, K+, and EC at the Beijing urban site did not influence the main content and conclusions of this paper, because the primary organic aerosols and secondary

inorganic aerosols were mainly discussed in this study.

 It is better to have discrimination of the ion concentration between daytime and nighttime for Figure S2 and Figure 2. In addition, the variation trend of CO concentration in Figure 2 may needs to be put together with gaseous pollutants such as SO2 and O3.

Response: According to the referee's suggestion, we added the comparison of chemical species concentrations between the daytime and nighttime, also we put the CO line with gaseous pollutants together. Moreover, the figure showing variation trends of gaseous pollutants were moved to the Supplementary Information (Figure S2), as has mentioned in question 1.

 The specific calculation formula of light absorption cross section needs to be provided in methodology section.

Response: As the reviewer suggested, we provided the calculation formula of light absorption cross sections in the methodology section.

Please refer to Line 211-222 :

After running the Mie calculation, the attenuation efficiency ( $Q_{atn}$ ), scattering efficiency ( $Q_{sca}$ ), and absorption efficiency ( $Q_{abs}$ ) of an individual particle were output with their definitions as follows (Aden and Kerker, 1951; Toon and Ackerman, 1981):

$$Q_{\rm atn} = \left(\frac{2}{x^2}\right) \sum_{n=1}^{\infty} (2n+1) [{\rm Re}(a_n+b_n)]$$
 (1)

$$Q_{\rm sca} = \left(\frac{2}{x^2}\right) \sum_{n=1}^{\infty} (2n+1)(|a_n|^2 + |b_n|^2)$$
(2)

$$Q_{\rm abs} = Q_{\rm atn} - Q_{\rm sca} \tag{3}$$

where  $x = \frac{\pi D}{\lambda}$ , is the dimensionless size parameter of the particle diameter *D* and the wavelength of light  $\lambda$ ;  $a_n$  and  $b_n$  are calculated from Riccati–Bessel functions of the particles sizes and refractive indices (Bohren and Huffman, 1983); The symbol Re denotes the real part of the complex quantity  $a_n+b_n$ .

The ACS of a particle can be obtained via multiplying the  $Q_{abs}$  by the geometric cross section of the particle as follow:

$$ACS = Q_{abs} \times \frac{\pi D^2}{4}$$
(4)

References:

Aden, A. L., and Kerker, M.: Scattering of Electromagnetic Waves from Two Concentric Spheres, Journal of Applied Physics, 22, 1242-1246, 10.1063/1.1699834, 1951.

Toon, O. B., and Ackerman, T. P.: Algorithms for the calculation of scattering by stratified spheres, Appl. Opt., 20, 3657-3660, 10.1364/ao.20.003657, 1981. Bohren, C. F., and Huffman, D. R.: Absorption and scattering of light by small

particles, John Wiley & Sons, New York, 1983.

10. Figure 9. Legend error. Ratio should be replaced by Eabs.

Response: Corrected.

---

## Author Comment (AC2) · 1 Jan 2021

**Reply to Referee#2:**

General Comments:

This study describes and compares wintertime aerosol particles in a rural location on the North China Plain with aerosol in Beijing (downwind from the rural site): aerosol concentration, chemical composition and individual-particle properties were measured, and optical properties simulated. The measurement campaign covered a period when conditions changed from relatively clean to heavily polluted, and the aerosol at the assumed source (the rural site) and in the urban haze could be compared. The major finding is that refractory primary organic particles from coal and biomass burning dominate at the rural site, and similar but coated particles are the major constituents in the aged haze. The inorganic coatings change the optical properties of the aerosol, by enhancing their absorption. The paper is sound and fairly well written. I have three concerns, one of which is partly a terminological issue.

Response: We thank referee#2' careful reading and critical comments which are helpful for improving the quality of our paper. All the comments and concerns raised by the referee have been explicitly considered and incorporated into the revised manuscript marked in red color.

Specific Comments:

1. Primary Organic Tar (POT). I am slightly uncomfortable with the use of the term "primary organic tar" (and its acronym, POT) for the observed class of particles. From the results it seems a correct interpretation that these particles are primary organics emitted by burning of coal and biomass. However, this is an interpretation and not an observation, as is now presented right at the start of the paper. The properties we know from TEM include the morphologies of the particles (some spherical and others irregular), their C-rich nature (although the particles were collected on a thick carbon+plastic film and therefore we do not know how much C they contain), and their thickness from SEM which suggests they were quite viscous when they arrived onto the grids. Indeed, these properties and the high concentrations at the locality where burning of coal is a major source of particles suggests they are primary organics. However, I am not sure they can be called "tar", since there is no study of the speciation of their components (although the authors refer to Xu et al. 2019 who used AMS and identified three groups of burning-related factors). Admittedly, "tar" is a vaguely defined substance. However, the atmospheric chemistry literature is already replete with vaguely defined terms (and acronyms), and introducing a new one, such as POT may not help to clear the confusion. For the spherical types of such particles already exists a widely used term, "tar ball". And about tar balls we know they are primary (Tóth et al. ACP 14, 6669-6675, 2014), indeed made of tar (Tóth et al. ACP 18, 10407–10418, 2018), and even their optical properties are known (Hoffer et al. ACP 16, 239-246, 2016). By using POT, I feel the definition of tar ball is diluted and thereby the utility of the term diminished. In summary, I believe it may be more prudent to call the POT particles just burning-related primary organics (PO), and it could be mentioned that the spherical ones appear to be tar balls. But, in the end, it is up to the authors what they wish to call these particles. Response: We appreciated the referee's critical comments. We accepted the referee's suggestion, the term POT was removed and burning-related primary organic aerosol (POA) particles were used in the revised manuscript, especially, the spherical primary organic particles were called tar balls as suggested.

2. Optical properties. The calculations for coated absorbing organic particles are interesting but I do not completely understand Figure 10. (1) What is the explanation for the insensitivity of ACS enhancement to the Dp/Dc size ratios larger than about 1.5? (2) What is the reason for the apparently random variation of ACS enhancement with Dc? I wonder if any optical measurements were done under hazy conditions during the sampling campaign? If such results are available, it would be interesting to compare them with the models.

Response: It will be very helpful to compare the simulation results with optical measurements. Unfortunately, the optical parameters were not measured during this observation, otherwise we would be willing to compare for further verification. (1) In Figure 10 we show that the absorption enhancement strongly depends on the core diameter and very little on the  $D_p/D_c$  when  $D_p/D_c>1.5$ . Similar variation trends were also reported in other studies (e.g., Bond et al., *J. Geophys. Res.*, 111, D20211, 2006; Wu et al., *Atmos. Chem. Phys.*, 18, 289–309, 2018). The "lensing effect" of secondary aerosol coating could significantly enhance the absorption capacity of the light-absorbing core. The incident light that would not pass through the core can finally reach the core and be absorbed after the refraction and internal reflections by the coating, which hence increases the

absorption cross sections. As has been illustrated in detail in the previous paper (Fuller et al., *J. Geophys. Res.*, 104, 15941–15954, 1999), the absorption cross sections rise monotonically as the thickness of the coating increases until the dipole resonance of the coated particle (for example  $D_p/D_c=1.5$  in this study) is reached. From there, the absorption cross sections oscillate more and more rapidly as higher-order particle resonances are encountered. (2) The ACS enhancement with  $D_c$  is not randomly varied. Enhancement is always greater than one—that is, coating always increases absorption, never decreases it; enhancement can be extremely high for small cores; and for larger cores enhancement is nearly constant. This phenomenon is introduced in detail in the previous paper which conducted the Mie calculations for the core-shell light-absorbing carbon particles (Bond et al., *J. Geophys. Res.*, 111, D20211, 2006).

Figure 1, showing the relationship among the absorption amplification (i.e., ACS enhancement in this study), core diameter, shell diameter, and  $D_p/Dc$ , was modified from this paper. It is clear that the small cores have very high enhancements; when the core size increases, the enhancement decreases largely and then after a certain core size the enhancement increases slowly but not beyond a certain value. Because the Mie calculations involve very complex electromagnetic theories and formulas, it is very hard to explain these results comprehensively. In this study, we used the results from the Mie calculation directly but do not go deep into the reasons for variations.

Figure 1. A contour plot of absorption amplification calculated by the concentric core-shell soot model at 550 nm. Assumptions: core m=1.85-0.71i and shell  $m=1.55-1.0^{-6}i$ . This graph is modified from Bond et al. (J. Geophys. Res., 111, D20211, 2006)

References:

Bond, T. C., Habib, G., and Bergstrom, R. W.: Limitations in the enhancement of visible light absorption due to mixing state, J. Geophys. Res.-Atmos., 111, https://doi.org/10.1029/2006JD007315, 2006.

Wu, C., Wu, D., and Yu, J. Z.: Quantifying black carbon light absorption enhancement with a novel statistical approach, Atmos. Chem. Phys., 18, 289-309, 10.5194/acp-18-289-2018, 2018.

Fuller, K. A., Malm, W. C., and Kreidenweis, S. M.: Effects of mixing on extinction by carbonaceous particles, J. Geophys. Res.-Atmos., 104, 15941-15954, https://doi.org/10.1029/1998JD100069, 1999.

3. Size changes. If the particles were transported from the rural site to Beijing, and during transport secondary aerosol formed on them, it does not see logical that their sizes decreased. The explanation offered is that large particles deposited but this process does not seem feasible for the particles in this size range. Can you please provide some more discussion on this issue?

Response: We are sorry for the puzzling problem. We should emphasize that the sizes of the whole aged particles indeed increased after the secondary aerosol formed on them as shown in Figure 7 in the manuscript. What we refer to is that the sizes of uncoated POA particles slightly decreased during the regional transport.

4. Line 29: "average particle-to-core ratios."- it is unclear what ratios; probably "diameter ratios" would be correct.

Response: We corrected it as "average particle-to-core diameter ratios".

- Line 32: replace "we using.." with "Using Mie theory we estimated.. " Response: Changed.
- 6. Line 61: chromatography Response: Corrected.
- 7. Line 64-66: subject of sentence unclear

Response: We revised this sentence in the revised manuscript as follows:

Line 65-67: Therefore, to document the aging processes of different particles in the NCP is of great significance for revealing the particle transformation in the atmosphere and better assessing the aerosol climatic effects.

- Line 85: "properties" instead of "information" Response: Changed.
- 9. Line 178-179: "five grids.. were selected" I guess "five grid meshes: : : were

selected" would be correct.

Response: Corrected.

10. Line 191: primary organic tar, POT: I do not understand the utility of this name and abbreviation. First, tar should be organic anyway, so "organic tar" is a tautology. Second, how do you know it is tar?

Response: As answered in question 1, we accepted the referee's suggestion and replaced the term POT with the primary organic aerosol (POA) particles, and the spherical organic particles were called tar balls.

11. Line 195-197: Hoffer et al. ACP 16, 239–246, 2016 found 1.84 - 0.21i for tar balls
- I wonder if this would make a large difference in your calculations, compared to the value by Alexander et al.

Response: We further calculated the absorption cross section and enhancement factors through Mie calculation using the refractive index of 1.84-0.21i, the results are listed in the Table below. There is no big difference between the results acquired by using the two refractive indices 1.67-0.27i and 1.84-0.21i.

Please see line 390-394:

It should be noted that another RI of 1.84-0.21i for tarballs was reported by Hoffer et al. (2016). The average Mie calculation results at the GC rural and BJ urban sites obtained by the RIs of 1.67-0.27i (used in this study) and 1.84-0.21i were compared and we found that the two RIs only cause little differences between the results (Table S3). Therefore, only the results from the RI of 1.67-0.27i were used and discussed in this study.

| Location | Pollution level      | $ACS_{actual}^{a}$ | ACS non-aging b | $E_{abs}{}^c$ | Refractive Index                               |
|----------|----------------------|--------------------|---------------------------------------|---------------|------------------------------------------------|
| Gucheng  | Heavy (Early stage)  | 3.09               | 3.01                                  | 1.02          |                                                |
|          | Heavy (Middle stage) | 3.97               | 3.53                                  | 1.12          | 1.67–0.27 i
(Alexander et al., 2008) |
|          | Heavy (Late stage)   | 4.43               | 3.18                                  | 1.39          |                                                |
| Beijing  | Moderate             | 2.06               | 1.86                                  | 1.10          |                                                |
|          | Heavy                | 3.00               | 2.15                                  | 1.39          |                                                |
| Gucheng  | Heavy (Early stage)  | 3.08               | 3.00                                  | 1.02          |                                                |
|          | Heavy (Middle stage) | 3.97               | 3.51                                  | 1.13          | 1.84–0.21 i
(Hoffer et al., 2016)    |
|          | Heavy (Late stage)   | 4.47               | 3.13                                  | 1.43          |                                                |
| Beijing  | Moderate             | 2.04               | 1.85                                  | 1.10          |                                                |
|          | Heavy                | 2.99               | 2.14                                  | 1.40          |                                                |

Table S3. Comparison between the average Mie calculation results acquired by two reported refractive indices of 1.67–0.27*i* and 1.84–0.21*i* in previous studies.

*a*ACSactual represents the absorption cross section of individual POA-containing particles (including core–shell POA–SIA and bare POA) under the actual scenario;

bACSnon-aging represents the absorption cross section of individual uncoated POA particles (including POA cores and bare POA) under the particle non-aging scenario;

 $^{c}E_{abs}$  represents the absorption enhancement factor due to the particle aging, i.e., ratio of ACSactual to ACSnon-aging.

References:

Alexander, D. T. L., Crozier, P. A., and Anderson, J. R.: Brown carbon spheres in East Asian outflow and their optical properties, Science, 321, 833-836, 2008.

Hoffer, A., Toth, A., Nyiro-Kosa, I., Posfai, M., Gelencser, A.: Light absorption properties of laboratory-generated tar ball particles, Atmos. Chem. Phys., 16, 239-246, 10.5194/acp-16-239-2016, 2016.

12. Line 246-249: unclear sentence, please reword

Response: We reworded this sentence in the revised manuscript as follows: Line 247-250: Contrasting to the transition periods at two sampling sites, we found that the SIA concentration increased significantly, meanwhile, the OM concentration only slightly increased at the GC rural and BJ urban sites with the consistent decreasing WS and increasing RH during the heavily polluted period.

13. Line 266-267 and text that follows: "TEM observations show a distinct group of spherical and irregular primary organic particles.." - I think TEM observations indeed show the morphologies of the particles but they do not provide direct information on their "organic" nature (only their elemental composition), let alone their primary origin. Please try to distinguish observations from interpretation.

Response: Thanks for this comment. According to the referee's suggestion, these sentences were modified to avoid over interpretation in the description of the observation results.

- 14. Line 278: Could it be just PO? Response: Accepted.
- 15. Line 295: "agrees" instead of "consists" Response: Corrected.
- 16. Line 372: influences Response: Corrected.
- 17. Line 376-377: how were the absorption cross sections estimated?Response: We added more description about the model calculations in the method

section 2.5. We think this will make the calculation results clearer.

Please see line 211-222:

After running the Mie calculation, the attenuation efficiency ( $Q_{atn}$ ), scattering efficiency ( $Q_{sca}$ ), and absorption efficiency ( $Q_{abs}$ ) of an individual particle were output with their definitions as follows (Aden and Kerker, 1951; Toon and Ackerman, 1981):

$$Q_{\rm atn} = \left(\frac{2}{x^2}\right) \sum_{n=1}^{\infty} (2n+1) [{\rm Re}(a_n+b_n)]$$
(1)

$$Q_{\rm sca} = \left(\frac{2}{x^2}\right) \sum_{n=1}^{\infty} (2n+1)(|a_n|^2 + |b_n|^2)$$
(2)

$$Q_{\rm abs} = Q_{\rm atn} - Q_{\rm sca} \tag{3}$$

where  $x = \frac{\pi D}{\lambda}$ , is the dimensionless size parameter of the particle diameter *D* and the wavelength of light  $\lambda$ ;  $a_n$  and  $b_n$  are calculated from Riccati–Bessel functions of the particles sizes and refractive indices (Bohren and Huffman, 1983); The symbol Re denotes the real part of the complex quantity  $a_n+b_n$ . The ACS of a particle can be obtained via multiplying the  $Q_{abs}$  by the geometric cross section of the particle as follows:

$$ACS = Q_{abs} \times \frac{\pi D^2}{4}$$
(4)

References:

Aden, A. L., and Kerker, M.: Scattering of Electromagnetic Waves from Two Concentric Spheres, J.Appl. Phys., 22, 1242-1246, 10.1063/1.1699834, 1951.

Toon, O. B., and Ackerman, T. P.: Algorithms for the calculation of scattering by stratified spheres, Appl. Opt., 20, 3657-3660, 10.1364/ao.20.003657, 1981.

Bohren, C. F., and Huffman, D. R.: Absorption and scattering of light by small particles, John Wiley & Sons, New York, 1983.

18. Line 405-409: this seems important, but I do not understand the statement completely.

Response: We are sorry for the puzzling issue. After consideration we decided to delete this sentence.